# ISG15 as a Potent Immune Adjuvant in MVA-Based Vaccines Against Zika Virus and SARS-CoV-2

**DOI:** 10.3390/vaccines13070696

**Published:** 2025-06-27

**Authors:** Juan García-Arriaza, Michela Falqui, Patricia Pérez, Rocío Coloma, Beatriz Perdiguero, Enrique Álvarez, Laura Marcos-Villar, David Astorgano, Irene Campaña-Gómez, Carlos Óscar S. Sorzano, Mariano Esteban, Carmen Elena Gómez, Susana Guerra

**Affiliations:** 1Department of Molecular and Cellular Biology, Centro Nacional de Biotecnología (CNB), Consejo Superior de Investigaciones Científicas (CSIC), 28049 Madrid, Spain; jfgarcia@cnb.csic.es (J.G.-A.); pperez@cnb.csic.es (P.P.); perdigue@cnb.csic.es (B.P.); enrique.alvarez@cnb.csic.es (E.Á.); lmarcos@cnb.csic.es (L.M.-V.); dastorgano@cnb.csic.es (D.A.); mesteban@cnb.csic.es (M.E.); cegomez@cnb.csic.es (C.E.G.); 2Centro de Investigación Biomédica en Red de Enfermedades Infecciosas (CIBERINFEC), Instituto de Salud Carlos III (ISCIII), 28029 Madrid, Spain; 3Department of Preventive Medicine, Public Health and Microbiology, Faculty of Medicine, Universidad Autónoma de Madrid, 28029 Madrid, Spain or michela.falqui@unive.it (M.F.); rcoloma@cnb.csic.es (R.C.); irene.campanna@uam.es (I.C.-G.); 4Biocomputing Unit and Computational Genomics, CNB, CSIC, 28049 Madrid, Spain; coss@cnb.csic.es; 5Department of Microbiology, Icahn School of Medicine at Mount Sinai, New York, NY 10029, USA; 6Global Health and Emerging Pathogens Institute, Icahn School of Medicine at Mount Sinai, New York, NY 10029, USA

**Keywords:** ISG15, adjuvant, vaccines, MVA, ZIKV, SARS-CoV-2, innate immunity, T-cell responses

## Abstract

**Background:** Vaccines represent one of the most affordable and efficient tools for controlling infectious diseases; however, the development of efficacious vaccines against complex pathogens remains a major challenge. Adjuvants play a relevant role in enhancing vaccine-induced immune responses. One such molecule is interferon-stimulated gene 15 (ISG15), a key modulator of antiviral immunity that acts both through ISGylation-dependent mechanisms and as a cytokine-like molecule. **Methods:** In this study, we assessed the immunostimulatory potential of ISG15 as an adjuvant in *Modified Vaccinia virus Ankara* (MVA)-based vaccine candidates targeting *Zika virus* (ZIKV) and *Severe Acute Respiratory Syndrome Coronavirus 2* (SARS-CoV-2). Early innate responses and immune cell infiltration were analyzed in immunized mice by flow cytometry and cytokine profiling. To elucidate the underlying mechanism of action of ISG15, in vitro co-infection studies were performed in macrophages. Finally, we evaluated the magnitude and functional quality of the elicited antigen-specific cellular immune responses in vivo. **Results:** Analysis of early innate responses revealed both platform- and variant-specific effects. ISG15AA preferentially promoted natural killer (NK) cell recruitment at the injection site, whereas ISG15GG enhanced myeloid cell infiltration in draining lymph nodes (DLNs), particularly when delivered via MVA. Moreover, in vitro co-infection of macrophages with MVA-based vaccine vectors and the ISG15AA mutant led to a marked increase in proinflammatory cytokine production, highlighting a dominant role for the extracellular, ISGylation-independent functions of ISG15 in shaping vaccine-induced immunity. Notably, co-infection of ISG15 with MVA-ZIKV and MVA-SARS-CoV-2 vaccine candidates enhanced the magnitude of antigen-specific immune responses in both vaccine models. **Conclusions:** ISG15, particularly in its ISGylation-deficient form, acts as a promising immunomodulatory adjuvant for viral vaccines, enhancing both innate and adaptive immune responses. Consistent with previous findings in the context of *Human Immunodeficiency virus type 1* (HIV-1) vaccines, this study further supports the potential of ISG15 as an effective adjuvant for vaccines targeting viral infections such as ZIKV and SARS-CoV-2.

## 1. Introduction

Vaccines are among the most effective interventions in global public health, representing a cost-efficient strategy to prevent infectious diseases and mitigate their associated morbidity and socioeconomic impact. The recent COVID-19 pandemic has shown the urgent need for rapid, versatile, and effective vaccine platforms to protect against emerging viral infections that can cause significant mortality and global economic impact. However, the development of effective vaccines poses significant scientific challenges. Many aspects of protective immunity remain unclear, highlighting the need for optimized formulations, including the rational use of adjuvants [1,2]. Adjuvants play an important role in optimizing vaccines. They are immunostimulatory components that enhance the magnitude, quality, and durability of antigen-specific immune responses to co-delivered antigens. Adjuvants can also reduce the antigen dose required to achieve protective immunity, a particularly relevant benefit for DNA-based vaccines, which typically require high doses in clinical trials [3,4,5].

Among candidate adjuvants, interferon-stimulated gene 15 (ISG15) has been identified as a promising immune modulator with dual antiviral and immunostimulatory functions. ISG15 is critically involved in the regulation of antiviral immunity by encoding a 15 kDa ubiquitin-like protein that can be covalently attached to target proteins through ISGylation, a post-translational modification that modulates multiple cellular and immune pathways and influences the outcome of viral infections [6]. Additionally, ISG15 can be secreted and functions as a cytokine-like molecule [7,8,9]. Extracellular ISG15 interacts with receptors on immune cells, such as lymphocyte function-associated antigen-1 (LFA-1) on natural killer (NK) cells, promoting the release of cytokines like IFN-γ and interleukin-10 (IL-10) [8]. It has also been implicated in the recruitment of IL-1β-producing CD8α+ dendritic cells (DCs) to the site of infection [10], underscoring its potential role as an immunomodulatory factor in vaccination [11,12].

In the context of vaccines, the *Modified Vaccinia virus Ankara* (MVA) is a widely used viral vector characterized by a well-established safety profile, which can accommodate a wide range of heterologous antigens, inducing strong immunogenicity and efficacy [13,14,15]. Due to its replication deficiency in human cells and ability to activate innate and adaptive cellular and humoral immunity, MVA has been employed as a vaccine vector against several pathogens in numerous preclinical and clinical studies [13,14,16]. MVA activates both Toll-like receptor (TLR)-dependent and -independent innate signaling pathways and promotes antigen presentation by DCs to stimulate adaptive immunity [17]. In heterologous prime/boost vaccination strategies, MVA is frequently used as a booster following priming with DNA, mRNA, or other viral vectors, a combination that has been shown to enhance the magnitude and quality of antigen-specific immune responses against several pathogens, such as *Human Immunodeficiency virus type 1* (HIV-1), *Hepatitis C virus*, *Human papillomavirus*, or *Severe Acute Respiratory Syndrome Coronavirus 2* (SARS-CoV-2) [18,19]. Furthermore, the immunogenicity of MVA vectors can be further optimized by deleting viral immunomodulatory genes or by co-expressing immune-stimulatory molecules, including cytokines and novel adjuvants [20,21,22,23].

Previous studies from our group have demonstrated the adjuvant properties of ISG15 in different immunization settings, including vaccines against HIV-1 and SARS-CoV-2 delivered via both DNA and MVA platforms [24,25]. In all cases, ISG15 significantly enhanced vaccine-induced immune responses, validating its role as a potent and key immunomodulatory adjuvant in the context of vaccination.

In this study, we used recombinant DNA and MVA vectors encoding either an ISGylation-competent form of ISG15 (ISG15GG) or an ISGylation-deficient mutant (ISG15AA), to dissect the impact of ISG15 on innate and adaptive immunity. We first assessed the immunological effects of DNA and MVA vectors expressing ISG15GG and ISG15AA variants in mice inoculated via the intramuscular route, focusing on innate immune cell recruitment. MVA-ISG15GG induced broad myeloid cell infiltration in lymphoid tissues, whereas MVA-ISG15AA preferentially promoted NK cell recruitment at the injection site. Both DNA-ISG15 constructs enhanced immune cell infiltration in muscle; however, only the secreted ISG15AA variant promoted cell recruitment to draining lymph nodes (DLNs). In vitro co-infection assays in macrophages further showed that ISG15AA triggered stronger proinflammatory cytokine responses, supporting the dominant role of ISGylation-independent, extracellular cytokine-like mechanisms in its adjuvant activity. Finally, we evaluated the immunostimulatory effects of ISG15GG and ISG15AA in mice immunized with MVA-based vaccine candidates expressing ZIKV prM-E antigens (MVA-ZIKV) or SARS-CoV-2 spike (S) antigen (MVA-Δ3-S). ISG15 expression enhanced antigen-specific immune responses in terms of magnitude in both vaccine models, with ISG15AA conferring superior immunogenicity in prime/boost protocols, thus supporting its potential as a next-generation genetic adjuvant.

## 2. Materials and Methods

### 2.1. Cells and Viruses

DF-1 chicken embryo fibroblast cells were cultured and maintained in Dulbecco’s Modified Eagle Medium (DMEM) (Lonza, Basel, Switzerland) supplemented with UltraGlutamine (Lonza), 4.5 g/L glucose (Lonza), 0.1 mM non-essential amino acids (Sigma-Aldrich, St. Louis, MO, USA), and 10% heat-inactivated fetal calf serum (FCS) (Sigma-Aldrich). HEK293T cells (ATCC catalog no. CRL-3216, Manassas, VA, USA) and immortalized murine bone marrow-derived macrophages (iBMDM) (kindly provided by Dr. María Ángeles Balboa (IBGM-CSIC, Valladolid, Spain) [26] were cultured in DMEM with 1 g/L glucose, supplemented with 0.1 mM non-essential amino acids and 10% heat-inactivated FCS.

The following recombinant MVA viruses were used: MVA-Δ3-GFP [24], MVA-B [27], MVA-ZIKV [28], MVA-Δ3-S [29], MVA-Δ3-ISG15GG [24], MVA-Δ3-ISG15AA [24], and MVA-WT. All heterologous genes were inserted into the MVA thymidine kinase (TK) locus. The encoded transgenes include green fluorescent protein (GFP); *gag-pol-nef* and env genes of HIV-1 clade B; *prM* and *E* genes of ZIKV; *S* gene of SARS-CoV-2; ISG15GG fused at its N-terminal end to the V5 tag, a polypeptide fragment derived from an epitope shared by the P and V proteins of simian virus 5 [30] (V5-ISG15GG); and ISG15AA fused to V5 at its N-terminal end (V5-ISG15AA). All recombinant viruses, except MVA-B and MVA-WT, were based on the MVA-Δ3 backbone, which includes targeted deletions of the *C6L*, *K7R*, and *A46R* immunomodulatory genes to enhance innate immune activation [24,31].

### 2.2. Development of Plasmids Expressing Secretion-Defective ISG15 Mutants

To explore the role of extracellular ISG15, we generated DNA plasmids encoding secretion-defective murine ISG15 (mISG15) variants. Based on previously described human ISG15 mutations that impair secretion (L72A, S83A, L85F) [32], equivalent murine point mutations (M70A, S81A, L83F) were identified by sequence alignment (Appendix A). We first constructed the DNA vectors pcDNA3-V5mISG15GG and pcDNA3-V5mISG15AA, encoding mature murine ISG15 (15 kDa) with an N-terminal V5 tag and either a C-terminal LRLRGG (ISG15GG) or LRLRAA (ISG15AA) motif. These constructs were subcloned from the original pCAGGS V5mISG15 (GG or AA) plasmids into the pcDNA3.2 vector. Secretion-defective point mutations were introduced into the ISG15GG and ISG15AA backbones using the Q5^®^ Site-Directed Mutagenesis Kit (New England Biolabs, Ipswich, MA, USA) following the manufacturer’s protocol, and employing the pcDNA3-V5mISG15 (GG or AA) plasmids as a template and specific oligonucleotides designed to generate the M70A, S81A, and L83F mutations. All constructs were verified by sequencing (Macrogen Inc., Seoul, Republic of Korea). Plasmid DNA was purified using the EndoFree Plasmid Mega Kit (QIAGEN GmbH, Hilden, NRW, Germany), following the manufacturer’s instructions.

### 2.3. Ethics Statement

All animal procedures were approved by the Ethical Committee of Animal Experimentation (CEEA) of CNB-CSIC and authorized by the Division of Animal Protection of the Comunidad de Madrid (PROEX 169.4/20). All procedures were conducted in full compliance with institutional guidelines, the European Directive 2010/63/EU, and Spanish legislation (Royal Decree RD 53/2013) for the protection of animals used for scientific purposes. Female C57BL/6JOlaHsd mice (6–8 weeks old) were purchased from Envigo Laboratories (Sant Feliu de Codines, Barcelona, Spain) and housed under specific pathogen-free (SPF) conditions at the CNB-CSIC animal facility (registration number ES280790000182). Efforts were undertaken to minimize animal pain and to reduce animal use to the necessary minimum.

### 2.4. Analysis of the Impact of ISG15 Variant Expression on Immune Cell Recruitment in Mice by Flow Cytometry

To evaluate the immunomodulatory effects of ISG15 variants (expressed from the MVA-Δ3 vector) on local and systemic immune cell recruitment, female C57BL/6JOlaHsd mice (6–8-week-old; n = 4 per group) were intramuscularly (i.m.) immunized with 1 × 10^7^ plaque-forming units (PFUs) of MVA-Δ3-ISG15GG (group 1), MVA-Δ3-ISG15AA (group 2), or MVA-Δ3-GFP (group 3). A PBS-treated group serves as a negative control (group 4). At day 1 post-inoculation, animals were sacrificed, and the injected muscle and inguinal DLNs were harvested and processed for flow cytometry analysis. Mechanical dissociation of muscle samples was followed by enzymatic digestion, following previously described protocols [24], while DLNs were passed through 70 μm cell strainers to obtain single-cell suspensions.

To evaluate the effect on immune cell recruitment of ISG15 variants expressed from plasmid DNA, an independent experiment was conducted with the same mouse strain and age (n = 4 per group). Mice were i.m. immunized with 50 μg of DNA-ISG15AA-Mut (a 1:1:1 mixture of plasmids pcDNA-ISG15AA M70A, pcDNA-ISG15AA S81A, and pcDNA-ISG15AA L83F) (group 1), wild-type pcDNA-ISG15-AA (group 2), or empty vector DNA-ϕ (group 3). PBS-treated mice were used as the negative control group (group 4). At day 4 post-inoculation, mice were sacrificed and the muscle tissues and the inguinal DLNs were collected and processed for flow cytometry analysis.

Flow cytometry analysis was conducted to assess immune cell recruitment to the muscle and DLNs after immunization with ISG15-based vectors, as previously described [24]. Briefly, 1 × 10^6^ cells per sample were plated in 96-well plates and incubated with LIVE/DEAD Fixable Red Dead Cell Stain Kit (Invitrogen, Carlsbad, CA, USA) at 4 °C for 30 min. Following Fc receptor blocking using anti-CD16/CD32 antibody (BD Biosciences, San Jose, CA, USA), cells were incubated with biotin-conjugated MHC-II (20 min, 4 °C). Then, cells were incubated with the following specific fluorochrome-conjugated surface antibodies for 15 min at 4 °C for the identification of various myeloid immune cell populations: Ly6G-PE, CD19-PE, CD3-PE, SiglecF-PE, Ly6C-PerCP, Avidin-PE-Cy7, CD64-APC, CD11b-Alexa Fluor 700, CD11c-APC-Cy7, CD45-Pacific Blue, and B220-BV510 (all from BD Biosciences). Samples were acquired on a Gallios flow cytometer (Beckman Coulter, Brea, CA, USA), and data were analyzed using FlowJo software (version 10.4.2; Tree Star, Ashland, OR, USA). Between 1 × 10^5^ and 5 × 10^5^ gated events were recorded per sample. Immune cell populations were identified based on previously established gating strategies [24].

### 2.5. Co-Infection Assays

For co-infection experiments, monolayers of iBMDM cells were infected at a multiplicity of infection (MOI) of 1 PFU/cell. In co-infections involving two different MVA recombinant viruses, each virus was added at an MOI of 0.5 PFUs/cell to maintain a total MOI of 1. After 1 h of viral adsorption at 37 °C, the inoculum was discarded, and cells were washed once with PBS. DMEM supplemented with 2% heat-inactivated FCS was then added, and cells were incubated for an additional 24 h. At the end of the incubation period, the supernatants were discarded, and cells were collected for downstream analysis, including Western blotting and quantitative real-time PCR (qPCR). Cell pellets were stored at −20 °C until use.

### 2.6. Protein and RNA Extraction and Quantification

For protein extraction, cells were lysed in 100 µL of Laemmli loading buffer [0.8% (*v*/*v*) sodium dodecyl sulfate (SDS) (VWR International, Radnor, PA, USA), 50 mM Tris(hydroxymethyl)aminomethane (Tris) (VWR International), 0.90 nM bromophenol blue (VWR International), and 30% (*v*/*v*) glycerol (Thermo Fisher Scientific, Waltham, MA, USA)]. Lysates were collected and kept at −20 °C until further use in Western blotting analysis. For RNA extraction, cells were lysed in 500 µL of NucleoZOL reagent (Macherey-Nagel GmbH & Co. KG, Düren, NRW, Germany) following the manufacturer’s instructions. Total RNA concentrations were measured using a MaestroNano spectrophotometer (MaestroGen Inc., Hsinchu City, Taiwan).

### 2.7. SDS-PAGE and Western Bloting Analysis

To analyze ISG15 expression, cell lysates were resuspended in Laemmli 1X (containing 2% SDS, 10% glycerol, 60 mM Tris-Cl pH 6.8, 0.01% Bromophenol Blue, and 20 mM dithiothreitol) and subsequently separated by 12% Sodium Dodecyl Sulfate Polyacrylamide Gel Electrophoresis (SDS-PAGE). Proteins were transferred to membranes and analyzed by Western blot using a monoclonal rabbit V5 tag-specific antibody (Cell signaling, Danvers, MA, USA, #13202). Horseradish peroxidase (HRP)-conjugated secondary antibodies (goat anti-rabbit, anti-mouse, or anti-Armenian hamster) (Sigma-Aldrich) were used as appropriate. Immune complexes were visualized using the Clarity Western ECL substrate (Bio-Rad, Hercules, CA, USA), and signal detection was performed with the ChemiDoc imaging system (Bio-Rad).

### 2.8. Reverse Transcription and Quantitative PCR (RT-qPCR)

Total RNA was converted into complementary DNA (cDNA) using the High-Capacity cDNA Reverse Transcription Kit (Applied Biosystems, Carlsbad, CA, USA). Briefly, 10 µL of extracted RNA was combined with 10 µL of a 2× RT master mix containing MultiScribe™ reverse transcriptase (5 U/µL), RT buffer, 8 mM deoxyribonucleotide triphosphates (dNTPs), random primers, and 2 U/µL RNase inhibitors (Hoffmann-La Roche, Basel, Switzerland), diluted in DEPC-treated water. RNA was reverse-transcribed into cDNA using an MJ Mini 48-well thermocycler (Bio-Rad) under the following cycling conditions: 25 °C for 10 min, 37 °C for 2 h, and 85 °C for 5 min. The resulting cDNA samples were stored at 4 °C until use.

For qPCR, the FastGene^®^ IC Green kit (NIPPON Genetics EUROPE, Düren, Germany) was used. A total of 100 ng of cDNA was added to a qPCR reaction mix containing FastGene^®^ IC Green and 400 nM of forward and reverse primers, all diluted in DEPC-treated water. Primer sequences will be provided upon request. Then, cDNA was amplified on a StepOnePlus™ qPCR system (Applied Biosystems) under the following conditions: initial denaturation at 95 °C for 2 min, followed by 40 amplification cycles consisting of denaturation at 95 °C for 5 s and annealing/extension at 65 °C for 30 s.

Cycle threshold (Ct) values were defined as the cycle at which the fluorescence signal exceeded a predetermined threshold. Relative gene expression levels were calculated using the 2^−ΔΔCt^ method. Briefly, ΔCt values were calculated by subtracting the Ct of the reference gene from that of the target gene, and ΔΔCt values were calculated by comparing ΔCt values from experimental and control conditions.

### 2.9. Immunization Protocols and Evaluation of Vaccine-Induced Immune Responses in Mice

#### 2.9.1. ZIKV Vaccine Immunization and Immune Response Assessment

To evaluate the adjuvant effect of ISG15 variants in MVA-ZIKV vaccination, female C57BL/6JOlaHsd mice (6–8 weeks old; n = 4 per group) (Envigo Laboratories) were immunized using a heterologous DNA/MVA prime/boost protocol. Mice were primed i.m. with 50 μg of a DNA construct encoding the prM-E antigens of ZIKV (DNA-ZIKV) [28], and co-administered with 50 μg of plasmids expressing ISG15GG or ISG15AA (DNA-ISG15GG and DNA-ISG15AA, respectively) or 50 μg of an empty DNA (DNA-Ф). Control mice were inoculated with a total amount of 100 μg of empty DNA (50 µg DNA-Ф + 50 µg DNA-Ф per mouse). DNA was administered bilaterally in the quadriceps (50 μL/leg). Twenty-two weeks later, mice were boosted intraperitoneally (i.p.) with 1 × 10^7^ PFUs of MVA-ZIKV expressing ZIKV prM and E antigens [28], while control mice were boosted i.p. with 1 × 10^7^ PFUs of MVA-WT. At 10 days post-boost, mice were sacrificed using CO_2_ and spleens were extracted and processed to assess ZIKV E-specific T-cell immune responses.

ZIKV E-specific CD4 and CD8 T-cell responses were assessed by intracellular cytokine staining (ICS) following stimulation of splenocytes with a peptide pool derived from the ZIKV E protein of the PRVABC59 strain (5 μg/mL; BEI Resources, National Institute of Allergy and Infectious Disease, National Institutes of Health, Bethesda, MD, USA). The peptide pool spans the full-length ZIKV E protein as overlapping 15-mers peptides with 12 amino acid offsets, as previously described [28]. Briefly, after spleen processing, 4 × 10^6^ splenocytes (depleted of red blood cells) were seeded into 96-well plates and stimulated for 6 h in complete RPMI 1640 medium supplemented with 10% FCS, 1 μ/mL Golgiplug (BD Biosciences, Franklin Lakes, NJ, USA) to block cytokine secretion, 1× monensin (eBioscience, Carlsbad, CA, USA), anti-CD107a–fluorescein isothiocyanate (FITC) (BD Biosciences), and the ZIKV E peptide pool (5 μg/mL). Cells were then washed, stained for surface markers, fixed, and permeabilized using the Cytofix/Cytoperm kit (BD Biosciences). Intracellular cytokine staining was then performed using fluorochrome-conjugated antibodies. Dead cells were excluded using the LIVE/DEAD Fixable Violet Dead Cell Stain Kit (Invitrogen). The antibodies used for functional analysis included CD3-phycoerythrin (PE)-CF594, CD4-allophycocyanin (APC)-Cy7, CD8-V500, IFN-γ–PE-Cy7, TNF-α–PE, and IL-2–APC, all from BD Biosciences.

CD4 Tfh cell responses against the ZIKV E antigen were analyzed by ICS, as previously described (60), and following the same approach used for the analysis of ZIKV E-specific CD4 and CD8 T cells. Splenocytes (4 × 10^6^ cells) were stimulated for 6 h in complete RPMI 1640 medium supplemented with 10% FCS, 1 μL/mL Golgiplug (BD Biosciences) to inhibit cytokine secretion, 1X monensin (eBioscience), anti-CD154 (CD40L)–PE (BD Biosciences), and a combination of ZIKV E protein (5 μg/mL) and ZIKV E peptide pool (5 μg/mL). Cells were then processed following the same protocol used for the analysis of ZIKV E-specific CD4 and CD8 T-cell responses. For functional analysis, the following fluorochrome-conjugated antibodies were used: CD154-PE, IL-4-Alexa Fluor 488, IL-21-APC, and IFN-γ-PE-Cy7. For phenotypic characterization, antibodies included CD4-Alexa Fluor 700, CD8-V500, PD-1 (CD279)-APC-efluor780, CXCR5-PECF594, and CD44-PE-Cy5 (SPRD). All antibodies were obtained from BD Biosciences.

Cells were acquired using a Gallios flow cytometer (Beckman Coulter, Brea, CA, USA), and data were analyzed with FlowJo software (version 10.4.2; Tree Star, Ashland, OR, USA).

#### 2.9.2. SARS-CoV-2 Vaccine Immunization and Immune Response Assessment

To analyze the impact of ISG15 in MVA-Δ3-S vaccination against SARS-CoV-2, four groups of female C57BL/6JOlaHsd mice (6–8 weeks old; n = 5/group) were immunized i.m. with 5 × 10^6^ PFUs of MVA-Δ3-S co-administered with 5 × 10^6^ PFUs of MVA-Δ3 expressing ISG15GG (MVA-Δ3-ISG15GG), ISG15AA (MVA-Δ3-ISG15AA), or GFP (MVA-Δ3-GFP) as a control group. MVA-Δ3-S encodes a full-length native non-stabilized SARS-CoV-2 antigen [29]. Immunizations were administered on days 0 (prime) and 14 (boost). At day 10 post-boost, splenocytes were isolated and stimulated ex vivo with peptide pools spanning SARS-CoV-2 S protein. The magnitude, breadth, and polyfunctionality of SARS-CoV-2 S-specific CD4 and CD8 T-cell responses were assessed by ICS, as previously described [29], and similar to the process described above.

### 2.10. Data Analysis and Statistics

Flow cytometry data were analyzed using FlowJo software (version 10.4.2, Tree Star, Ashland, OR, USA). To assess the recruitment of immune cell populations, an ordinary one-way analysis of variance (ANOVA) with multiple comparisons was applied to determine differences between groups. For qPCR data, statistical analysis was also performed using one-way ANOVA followed by Tukey’s honest significant difference (HSD) test. Statistical analysis of T-cell immunogenicity data obtained by ICS was performed using a background-corrected approach that calculates confidence intervals and *p*-values, as previously described [24]. All statistical analyses and graphical representations were performed using GraphPad Prism 10.1.0 software. Statistical significance is denoted as follows: *, *p* < 0.05; **, *p* < 0.005; ***, *p* < 0.001.

## 3. Results

### 3.1. Platform-Dependent Modulation of Early Innate Immunity by ISG15 Variants

To assess the impact of ISG15 variants on early innate immune activation, we first analyzed immune cell infiltration in muscle tissue and DLNs following MVA-based vaccination. Four groups of female C57BL/6JOlaHsd mice were i.m. injected with 1 × 10^7^ PFUs of either MVA-Δ3-ISG15GG (G1), MVA-Δ3-ISG15AA (G2), MVA-Δ3-GFP (G3), or PBS as a control (G4). At 24 h post-injection, muscle samples from the injection site and ipsilateral inguinal DLNs were harvested and processed for flow cytometry analysis.

In vivo analysis revealed distinct patterns of immune cell infiltration among the experimental groups (Figure 1). In the muscle (Figure 1A), all MVA-treated mice showed increased recruitment of innate immune cells compared to PBS controls, particularly Ly6C^high^ monocytes, eosinophils, and neutrophils. MVA-Δ3-ISG15GG and MVA-Δ3-GFP induced comparable levels of inflammatory Ly6C^high^ monocytes, while MVA-Δ3-ISG15AA elicited slightly lower levels of these cells and a more pronounced reduction in eosinophils and neutrophils. However, MVA-Δ3-ISG15AA promoted a trend toward higher accumulation of NK cells than the other groups, suggesting a qualitatively distinct innate profile. In DLNs (Figure 1B), all MVA-treated groups also showed elevated cellular infiltration compared to the PBS control group, confirming early lymphoid activation. Among them, MVA-Δ3-ISG15GG induced the highest levels of Ly6C^high^ monocytes and neutrophils, followed by MVA-Δ3-GFP. In contrast, MVA-Δ3-ISG15AA showed reduced recruitment of these subsets. DC populations, including plasmacytoid DCs (pDCs), conventional DCs (cDC1 and cDC2), and migratory DCs (migDCs), were similarly enriched in the MVA-Δ3-ISG15GG and MVA-Δ3-GFP groups, but diminished in the MVA-Δ3-ISG15AA group. These findings suggest that MVA-Δ3-ISG15GG enhances broad myeloid cell trafficking to lymphoid tissue, while MVA-Δ3-ISG15AA promotes a more restricted innate response, characterized by preferential NK cell enrichment at the injection site and reduced granulocyte and DC recruitment.

Additionally, to assess the impact of the delivery platform and ISG15 secretion, we performed parallel experiments using ISG15 variants delivered via plasmid DNA. Murine point mutations equivalent to known secretion-defective human ISG15 variants (M70A, S81A, L83F) were identified by sequence alignment (Appendix A), and DNA vectors (ISG15GG or ISG15AA) encoding the mutant forms M70A, S81A, and L83F were generated following the protocol detailed in Materials and Methods. Then, HEK-293 cells were co-transfected with each DNA construct with or without plasmids encoding the ISGylation machinery (E1, E2, and E3). Expression of both wild-type and mutant ISG15 variants was confirmed by Western blotting using antibodies specific to the V5 epitope tag (Appendix A). To evaluate the in vivo effects of these DNA-based constructs, four groups of female C57BL/6JOlaHsd mice were immunized i.m. with 50 μg of a 1:1:1 mixture of plasmid DNA encoding either the secretion-defective ISG15AA mutant (DNA-ISG15AA-Mut; G1), wild-type ISG15AA (DNA-ISG15AA; G2), or an empty plasmid (DNA-ϕ; G3). A PBS-injected group served as a negative control (G4). On day 4 post-injection, muscle and DLNs were collected for flow cytometry. In the muscle (Figure 2A), both DNA-ISG15AA and DNA-ISG15AA-Mut significantly increased the levels of NK cells, Ly6C^high^ monocytes, and migDCs compared to DNA-ϕ and PBS controls, with no major differences between both groups, indicating that secretion of ISG15 is not required to enhance local innate immune infiltration when delivered via DNA. In contrast, in the DLNs (Figure 2B), the absence of ISG15 secretion in the DNA-ISG15AA-Mut group was associated with a marked reduction in several immune cell populations, particularly Ly6C^high^ monocytes and cDCs, compared to DNA-ISG15AA. These findings demonstrate that ISG15 secretion is essential for effective immune cell recruitment to lymphoid tissues in the context of plasmid DNA-based delivery.

Taken together, these findings indicate that the immune-stimulatory adjuvant effects of ISG15AA are platform-dependent. While local innate immune activation in muscle occurs irrespective of secretion, ISG15 secretion is essential for effective recruitment and activation of immune cells in the DLNs when delivered via DNA.

### 3.2. MVA-Δ3-ISG15AA Enhances Inflammatory Cytokine Responses During Co-Infection with MVA-Based Vaccine Vectors

Macrophages critically coordinate the interplay between innate and adaptive immunity, making them a relevant model to evaluate the immunomodulatory effects of ISG15 overexpression [33,34]. Thus, to investigate the capacity of ISG15 variants to enhance innate responses, we performed in vitro co-infection assays using iBMDMs and recombinant MVA vectors encoding antigens from relevant viral pathogens. We tested combinations of MVA vectors encoding antigens from HIV-1 (MVA-B), ZIKV (MVA-ZIKV), or SARS-CoV-2 (MVA-Δ3-S), along with vectors expressing either wild-type murine ISG15 (MVA-Δ3-ISG15GG) or its ISGylation-deficient mutant (MVA-Δ3-ISG15AA).

Thus, iBMDMs were infected with the different MVA-based vaccine vectors (MVA-B, MVA-ZIKV, and MVA-Δ3-S), either individually or co-infected with MVA-Δ3-ISG15GG or MVA-Δ3-ISG15AA at a combined MOI of 1 PFU/cell (0.5 PFUs per virus), and cytokine expression was assessed 24 h post-infection by RT-qPCR. MVA-B served as a control vector, as previous in vitro studies have demonstrated that co-infection of ISG15, in both GG and AA forms, does not interfere with antigen expression from this construct [24,35]. As shown in Figure 3, all recombinant MVA vectors upregulated expression of pro-inflammatory cytokines (IL-6, type I IFN) and ISG15 relative to mock-infected controls. Among them, co-infection with MVA-Δ3-ISG15AA consistently induced the highest cytokine expression, particularly for IL-6 and IFN-I, regardless of the co-infecting MVA-based antigen vector. Increases in ISG15 mRNA levels were moderate and, although not always statistically significant compared to the ISG15GG variant or vector alone, trends suggested a more potent proinflammatory profile associated with ISG15AA. Additionally, a slight reduction in IFN-I expression was observed in co-infections of MVA-Δ3-S or MVA-ZIKV with MVA-Δ3-ISG15AA compared to the co-infection of MVA-B with MVA-Δ3-ISG15AA, potentially reflecting vector-specific modulation.

In summary, these data demonstrate that overexpression of ISG15, particularly the non-conjugatable ISG15AA variant, amplifies proinflammatory cytokine production in macrophages when co-delivered with vaccine antigens. The results suggest that extracellular, cytokine-like functions of ISG15AA predominate in this context, highlighting its potential as an immune-stimulating adjuvant.

### 3.3. DNA-ISG15AA Co-Administration Enhances ZIKV-Specific CD4 and CD8 T-Cell Responses in Immunized Mice

After establishing the role of ISG15 secretion in modulating innate immune cell recruitment, we next evaluated its immunostimulatory potential as a genetic adjuvant in a heterologous DNA/MVA prime/boost vaccination regimen against ZIKV—an epidemiologically relevant virus in 2025, with ongoing transmission in parts of Latin America and Asia, and increasing concern in Europe due to climate-driven mosquito expansion [36].

Thus, to determine whether ISG15, in its ISGylation-competent (ISG15GG) or -deficient (ISG15AA) form, plays an important role as an adjuvant in the ZIKV-specific immune responses induced by MVA-ZIKV, we carried out an in vivo study in mice, as described in Materials and Methods. Thus, female C57BL/6JOlaHsd mice (6–8 weeks old, n = 4/group) were primed with plasmids encoding ISG15GG, ISG15AA, or an empty vector, co-delivered with DNA-ZIKV, and boosted with MVA-ZIKV 22 weeks later. Then, ZIKV E-specific T-cell responses were evaluated 10 days post-boost by ICS following ex vivo stimulation of splenocytes with ZIKV E overlapping peptide pools. The magnitude and quality of the T-cell responses were assessed by quantifying the frequency of IFN-γ, IL-2, TNF-α, and/or CD107a positive cells among CD4 and CD8 T cells.

As shown in Figure 4A, co-administration of DNA-ZIKV with DNA-ISG15GG or DNA-ISG15AA significantly enhanced ZIKV E-specific CD4 (left panel) and CD8 (right panel) T-cell responses compared to animals primed with only DNA-ZIKV. Notably, mice primed with DNA-ZIKV + DNA-ISG15AA induced the highest percentages of ZIKV E-specific CD4 and CD8 T cells secreting IFN-γ, IL-2, and TNF-α and expressing CD107a. Functional profiling revealed that all immunized mice induced highly polyfunctional ZIKV E-specific CD4 and CD8 T cells, with a greater proportion of cells co-expressing three or four functions (Figure 4B), but again, mice primed with DNA-ZIKV + DNA-ISG15GG and DNA-ZIKV + DNA-ISG15AA induced the highest percentages of those cells compared to the control group (DNA-ZIKV/MVA-ZIKV). In addition, co-administration of DNA-ZIKV with DNA-ISG15GG or DNA-ISG15AA significantly increased the frequency of ZIKV E-specific CD4 T follicular helper (Tfh) cells (PD-1^+^, CXCR5^+^) secreting IFNy and/or IL-21 and expressing CD40L, compared to regimens lacking ISG15 (Figure 4C). Furthermore, these ZIKV E-specific CD4 Tfh cells also exhibited enhanced polyfunctionality, as defined by the co-expression of IL-21, IFNy, and CD40L (Figure 4D). This functional enhancement was most pronounced in mice receiving ISG15 during the priming, suggesting that ISG15 promotes both the magnitude and the quality of the CD4 Tfh response.

In summary, these results reinforce our previous in vitro and in vivo findings on innate immune cell recruitment, demonstrating that DNA-mediated delivery of ISG15AA markedly enhances both the magnitude and the functional quality of ZIKV-specific CD4 and CD8 T-cell responses following MVA-ZIKV boost. Collectively, these data validate ISG15AA as a potent immunomodulatory adjuvant and support its use in future vaccine strategies targeting emerging viral pathogens.

### 3.4. Co-Administration of MVA-Δ3-ISG15 Enhances SARS-CoV-2-Specific T-Cell Responses Induced by MVA-Δ3-S Vaccination in Immunized Mice

Following the enhanced ZIKV-specific CD4 and CD8 T-cell responses observed with ISG15AA, we next evaluated whether similar immunopotentiation effects could be achieved in a different antigenic context, specifically vaccination with MVA-Δ3-S expressing the full-length native SARS-CoV-2 S protein. To this end, the adjuvant effect of ISG15 variants delivered via MVA vectors was evaluated in mice immunized with MVA-Δ3-S, as detailed in Materials and Methods. Mice received two doses (days 0 and 14) of MVA-Δ3-S with either MVA-Δ3-ISG15AA or MVA-Δ3-ISG15GG. Controls received MVA-Δ3-S + MVA-Δ3-GFP or MVA-Δ3-GFP alone. At 10 days post-boost, animals were sacrificed and splenocytes were obtained to evaluate S–specific CD4 and CD8 T-cell responses by ICS following ex vivo stimulation with SARS-CoV-2 S overlapping peptide pools.

As shown in Figure 5A, co-administration of MVA-Δ3-S with MVA vectors expressing ISG15 variants (ISG15GG or ISG15AA) significantly enhanced both S-specific CD4 (left panel) and CD8 (right panel) T-cell responses compared to MVA-Δ3-S + MVA-Δ3-GFP control group. The magnitude of the S-specific CD4 and CD8 T-cell responses were similar between the groups MVA-Δ3-S + MVA-Δ3-ISG15GG and MVA-Δ3-S + MVA-Δ3-ISG15AA (Figure 5A). 

Next, we examined the functional characteristics of the S-specific CD4 and CD8 T cells by assessing the expression of the degranulation marker CD107a and the production of cytokines IFN-γ, IL-2, and TNF-α (Figure 5B). Both MVA-Δ3-ISG15GG and MVA-Δ3-ISG15AA co-administration enhanced the quality of the T-cell response. Compared to the MVA-Δ3-S + MVA-Δ3-GFP group, the ISG15-adjuvanted groups showed a higher proportion of polyfunctional CD4 T cells, particularly those exhibiting three and four functions (Figure 5B, left panel). The enhancement was particularly evident for CD8 T cells (Figure 5B, right panel), where the MVA-Δ3-S + MVA-Δ3-ISG15GG and MVA-Δ3-S + MVA-Δ3-ISG15AA groups displayed markedly higher frequencies of highly polyfunctional cells, predominantly triple-positive for CD107a, IFN-γ, and TNF-α, compared to the MVA-Δ3-S + MVA-Δ3-GFP group.

We next investigated the potential impact of ISG15 on the activation of CD8 T cells targeting the MVA vector (Appendix A). To this end, we quantified VACV E3-specific CD8 T cells producing any combination of CD107a, IFN-γ, IL-2, and/or TNF. Co-administration of MVA-Δ3-S with MVA vectors expressing ISG15 variants (ISG15GG or ISG15AA) significantly enhanced VACV E3-specific CD8 T-cell responses compared to the MVA-Δ3-S + MVA-Δ3-GFP control group, but all of them induced lower VACV E3-specific CD8 T-cell responses compared to the homologous MVA-Δ3-GFP + MVA-Δ3-GFP control group (Appendix A). Moreover, Boolean analysis of CD8 T-cell polyfunctionality revealed that the MVA-Δ3-S + MVA-Δ3-ISG15GG and MVA-Δ3-S + MVA-Δ3-ISG15AA groups displayed markedly higher frequencies of highly polyfunctional cells, predominantly triple-positive for CD107a, IFN-γ, and TNF-α, compared to the MVA-Δ3-S + MVA-Δ3-GFP group (Appendix A).

While our study did not observe evidence of non-specific T-cell activation following ISG15AA adjuvantation, transient innate immune activation may occur and could potentially confer temporary protection against unrelated viral infections. This warrants further investigation, especially in the context of respiratory pathogens. In summary, these results demonstrate that co-administration of MVA-encoded ISG15, particularly the ISGylation-deficient form (ISG15AA), enhances the magnitude and quality of antigen-specific T-cell responses against SARS-CoV-2. These results support the broad immunostimulatory capacity of ISG15 across different antigens and vaccine platforms.

## 4. Discussion

The success of vaccination depends on eliciting a robust and appropriate immune response against a target antigen, while ensuring safety for human use [37]. In this study, we explored the immunomodulatory potential of ISG15, particularly the ISGylation-deficient mutant ISG15AA, as a genetic adjuvant delivered via MVA or plasmid DNA platforms. Our findings reveal a context-dependent role for ISG15 in shaping innate and adaptive immunity, reinforcing its relevance as a molecular adjuvant candidate. In accordance with WHO guidelines on the non-clinical evaluation of vaccine adjuvants (Annex 1, TRS No. 927), we provide data on the immunogenicity and local immune activation of ISG15 variants used as adjuvants. Future studies should further address pharmacokinetic and toxicological aspects, particularly for clinical translation.

Our results demonstrate that ISG15-mediated modulation of innate immune responses is strongly influenced by both the variant expressed (ISG15GG vs. ISG15AA) and the delivery platform (MVA vs. plasmid DNA). Direct comparisons between DNA- and MVA-based vaccine platforms are inherently limited by differences in vector biology and the timing of immune activation. In our study, samples were collected at 24 h post-infection for MVA and at day 4 post-injection for DNA vaccination, in line with each platform’s optimal time window for innate response analysis. These differences should be considered when interpreting cross-platform results. Consistent with previous studies identifying extracellular ISG15 as an endogenous damage-associated molecular pattern (DAMP) [6,8], we observed that ISG15AA enhances early NK cell and monocyte recruitment at the injection site (muscle). This effect was preserved even when ISG15 secretion was impaired in the DNA platform. However, recruitment of myeloid cells to DLNs was notably reduced with secretion-deficient ISG15AA, particularly in DNA-based immunization, highlighting the importance of extracellular ISG15 in propagating immune signals beyond the local site of injection. These observations align with previous reports showing that secreted ISG15 can act on DCs and macrophages in a paracrine fashion to promote antigen presentation and cytokine production [38]. Interestingly, MVA-Δ3-ISG15AA vaccination resulted in reduced infiltration of eosinophils and neutrophils at the injection site compared to other groups. This may suggest a lower local reactogenicity profile, although further studies are needed to confirm these observations and assess potential clinical relevance. In contrast, in the MVA setting, ISG15GG (which retains its conjugation activity) induced broader granulocyte and DC infiltration in lymphoid tissues, suggesting that ISGylation-dependent mechanisms may contribute to immune priming in this context.

Overall, our data support a model in which ISG15 exerts distinct immunostimulatory effects depending on the delivery platform, requiring secretion for lymphoid engagement in DNA-based vaccination, and involving both ISGylation and paracrine signaling in MVA-based vectors.

Notably, co-delivery of ISG15AA significantly enhanced antigen-specific T-cell responses in vivo. In the ZIKV vaccination models, DNA priming with ISG15AA followed by MVA-ZIKV boosting elicited the strongest CD4 and CD8 T-cell responses, with increased polyfunctionality. Similar enhancement was observed in the SARS-CoV-2 model, where co-administration of MVA-Δ3-ISG15AA or ISG15GG with MVA-Δ3-S improved the magnitude and quality of T-cell responses, particularly in triple-positive CD107a^+^IFN-γ^+^TNF-α^+^ subsets. These findings underscore the broad immunostimulatory potential of ISG15, with ISG15AA emerging as a particularly effective variant when delivered independent of the antigen. Regarding the humoral responses, while this study focused primarily on cellular immune responses, the effect of ISG15 on antibody-mediated responses remains to be fully elucidated. Interestingly, preliminary data from ongoing studies in SARS-CoV-2 and HIV models suggest that ISG15 does not substantially influence antibody production, which aligns with its role in promoting innate and cellular immune responses rather than directly enhancing B-cell activity. This further defines the adjuvant profile of ISG15 as skewing towards T-cell-mediated immunity.

From a translational perspective, ISG15AA may be a valuable component in heterologous prime/boost strategies or combinatorial vaccine formulations. In particular, ISG15 shows strong potential as an adjuvant in recombinant MVA-based vaccines. The use of dual-vector systems, separating antigen and adjuvant expression, offers a modular approach to fine-tuning immune responses. Our data suggest that this immunostimulatory effect is amplified by elevated extracellular ISG15 levels, positioning ISG15AA as a superior alternative to ISG15GG in this context. Although a single-vector system expressing both components could be advantageous, further development is required. The potential induction of anti-vector immunity with repeated administration of MVA-based vaccines warrants careful evaluation, as it may impact booster efficacy. The heterologous DNA/MVA prime/boost regimen used here may help mitigate this risk, but further studies are required to explore long-term vector immunity. The transition from in vitro studies to in vivo murine models has been crucial to assess the efficacy and safety of potential vaccines, especially considering the strong pro-inflammatory response that ISG15AA triggers in murine macrophages. Although ISG15AA expression increased proinflammatory cytokine production in macrophages, in vivo administration did not result in overt systemic inflammation or toxicity. Nonetheless, detailed toxicology assessments will be essential prior to clinical development. Also, future preclinical studies in other animal models such as non-human primates and clinical trials will determine whether ISG15 can be a useful adjuvant for vaccine immunization. Finally, although we did not perform specific cross-reactivity tests in human tissue cultures, this is an important consideration for vector-based vaccines. Future studies using in vitro tissue models will be essential to evaluate off-target effects and predict potential reactogenicity in humans. In conclusion, this work highlights ISG15, especially the ISGylation-deficient ISG15AA variant, as a potent and versatile genetic adjuvant capable of enhancing both innate and adaptive immune responses. These findings support the continued development of ISG15-adjuvanted vaccines, particularly in settings where robust cellular immunity is desired, and contribute to the growing toolkit of genetic adjuvants for next-generation vaccine design.

## 5. Conclusions

This study demonstrates that ISG15, particularly its ISGylation-deficient variant ISG15AA, acts as a potent and broadly applicable immune adjuvant, significantly enhancing vaccine-induced immune responses against ZIKV, SARS-CoV-2, and HIV-1. Delivered via MVA vectors or plasmid DNA, ISG15AA improves early innate immune cell recruitment, amplifies proinflammatory cytokine production (e.g., IFN-γ, IL-6, TNF-α), and enhances both the magnitude and the polyfunctionality of antigen-specific CD4 and CD8 T-cell responses.

Importantly, ISG15AA consistently outperforms ISG15GG, highlighting the dominant role of its extracellular, cytokine-like function over intracellular ISGylation. This enhanced effect was particularly evident in macrophage stimulation assays and in heterologous prime/boost vaccination settings.

These results position ISG15AA as a promising next-generation adjuvant for improving vaccines against emerging and persistent viral infections. Further in vivo studies will be essential to assess its long-term safety and translational potential.

## Figures and Tables

**Figure 1 vaccines-13-00696-f001:**
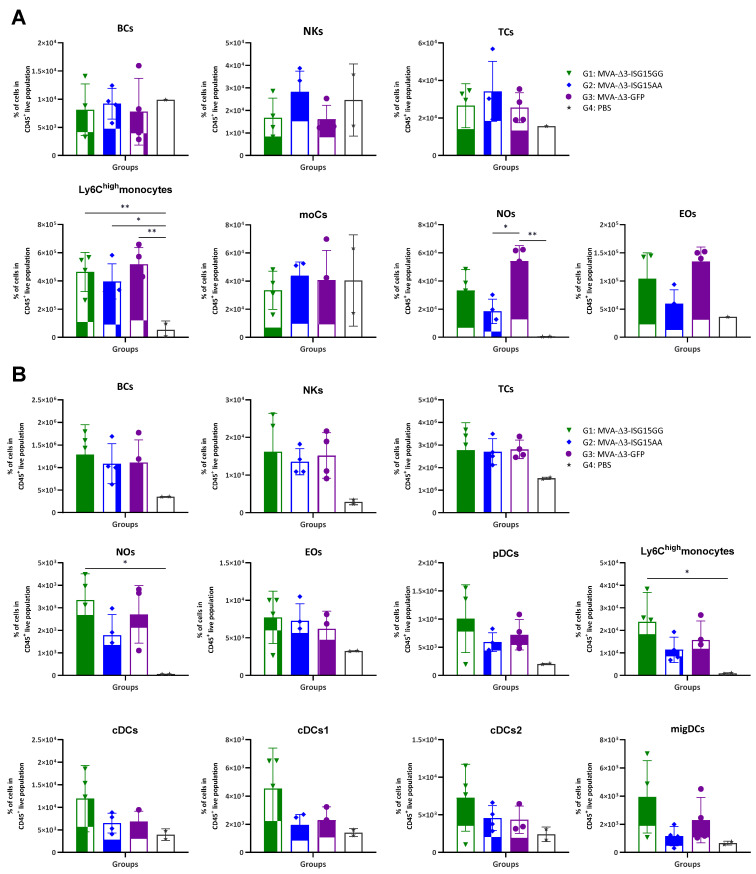
Recruitment of innate immune cells in muscle and DLNs following immunization with MVA-based vectors expressing ISG15 variants. Female C57BL/6 mice were immunized i.m. with MVA-Δ3-ISG15GG, MVA-Δ3-ISG15AA, or MVA-Δ3-GFP. PBS-injected mice served as controls. At 24 h post-injection, immune cell populations in the injected muscle (**A**) and inguinal DLNs (**B**) were analyzed by flow cytometry. Data are presented as individual values (colored symbols) with mean ± standard deviation (SD). Cell populations analyzed include BCs, B cells; NKs, natural killer cells; TCs, T cells; moCs, monocyte-derived cells; Nos, neutrophils; Eos, eosinophils; pDCs, plasmacytoid dendritic cells; cDCs, conventional dendritic cells; migDCs, migratory dendritic cells. *, *p* < 0.05; **, *p* < 0.005.

**Figure 2 vaccines-13-00696-f002:**
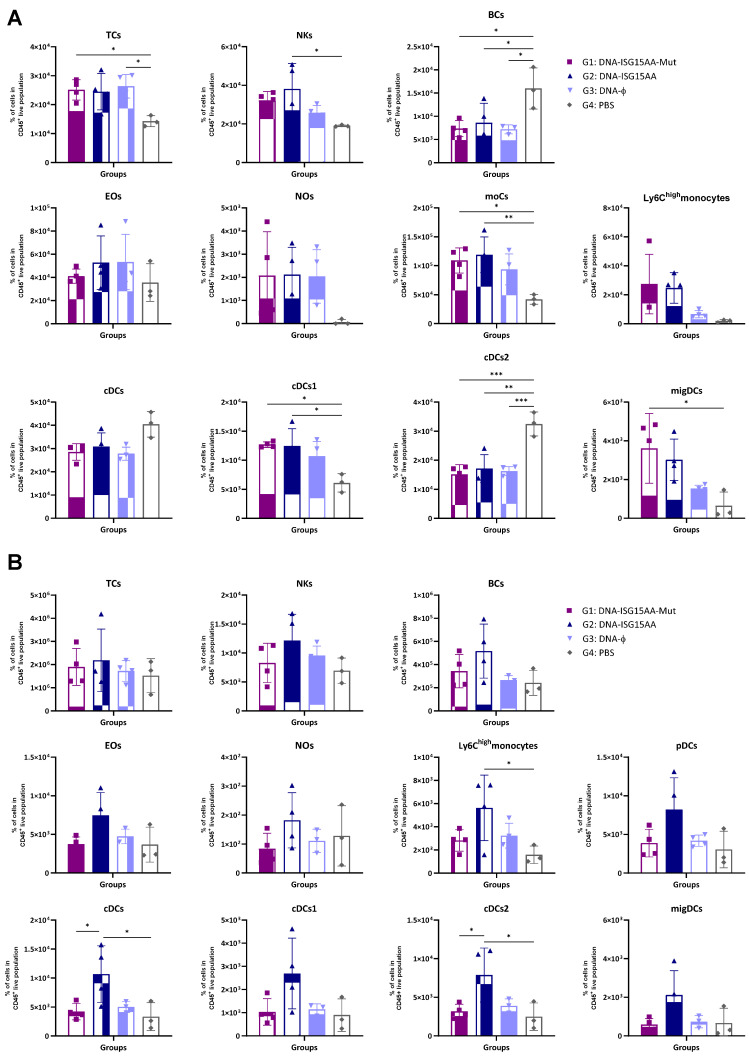
Recruitment of innate immune cells in muscle and DLNs following immunization with DNA-based vectors expressing ISG15 variants. Female C57BL/6 mice were immunized i.m. with plasmid DNA vectors encoding wild-type ISG15AA or a 1:1:1 mixture of secretion-deficient ISG15AA mutants (M70A, S81A, L83F). Immune cell subsets in muscle (**A**) and DLNs (**B**) were analyzed by flow cytometry 4 days after injection. Data are presented as mean ± SD. Cell subsets analyzed include TCs, T cells; NKs, natural killer cells; BCs, B cells; Eos, eosinophils; Nos, neutrophils; moCs, monocyte-derived cells; pDCs, plasmacytoid dendritic cells; cDCs, conventional dendritic cells; migDCs, migratory dendritic cells. *, *p* < 0.05; **, *p* < 0.005; ***, *p* < 0.001.

**Figure 3 vaccines-13-00696-f003:**
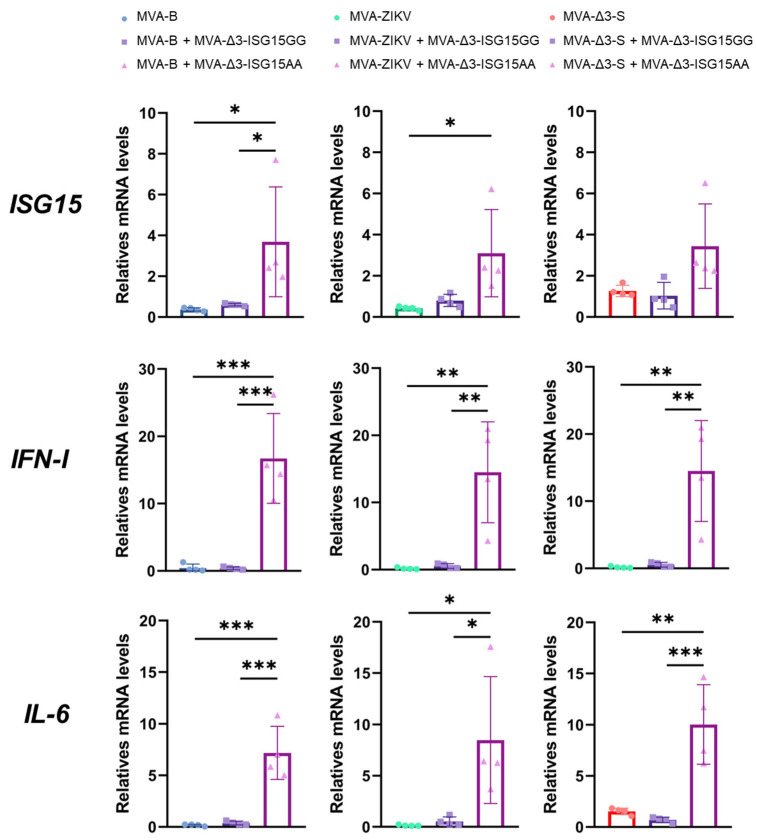
ISG15AA enhances cytokine production in co-infected macrophages. iBMDMs were infected or co-infected with MVA vectors encoding vaccine antigens from HIV-1, ZIKV or SARS-CoV-2, and ISG15 variants (ISG15GG or ISG15AA). RT-qPCR quantification of murine ISG15, IL-6, and IFN-I gene expression. Hypoxanthine phosphoribosyl transferase (HPRT) was used as the reference gene, and mRNA levels were measured in arbitrary units (A.U.) and normalized relative to uninfected cells (Mock). Comparison of quantification results between single infection with the vaccine virus and co-infections with the virus expressing the ISG15GG or ISG15AA adjuvants are shown. Data were organized in a panel based on the quantified cytokine and the vaccine virus used. Results were normalized to the parental virus (MVA-Δ3-GFP). Thick bars represent the mean of four replicates, and error bars indicate SD. *, *p* < 0.05; **, *p* < 0.01; ***, *p* < 0.001). To facilitate correct interpretation, the mean values obtained for the co-infection with MVA-Δ3-ISG15AA and MVA-B, MVA-ZIKV, or MVA-Δ3-S are provided below: ISG15: 3.68, 3.09, 3.44; IFN-I: 16.71, 14.49, 14.23; IL-6: 7.18, 8.48, 10.02.

**Figure 4 vaccines-13-00696-f004:**
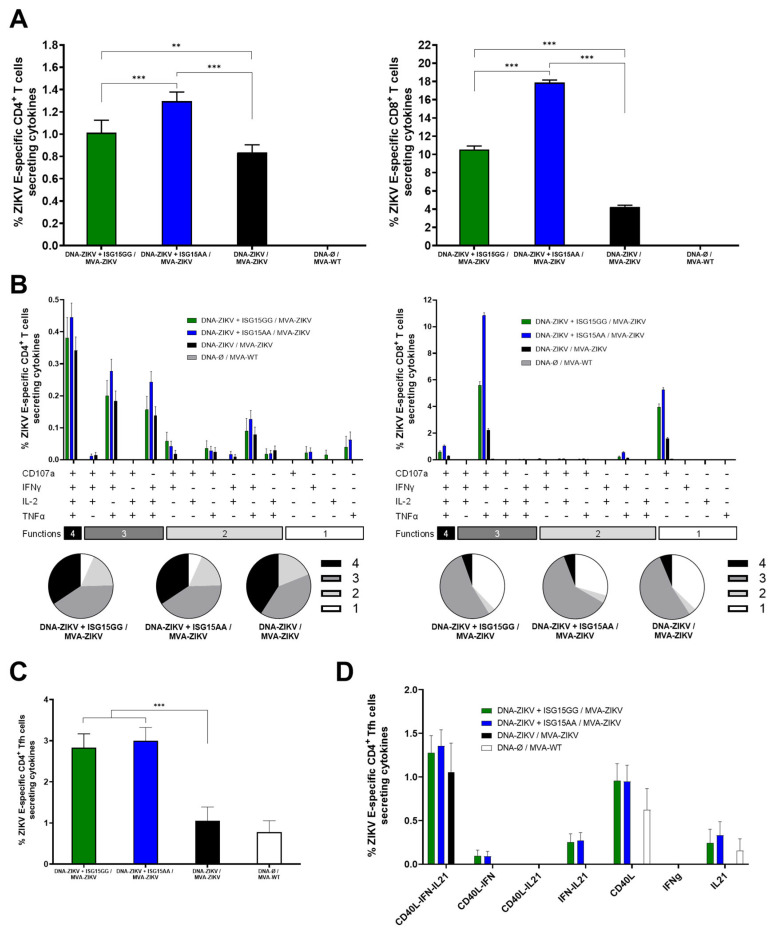
ISG15AA enhances ZIKV-specific T-cell responses in heterologous prime/boost vaccination in mice. Female C57BL/6 mice (n = 4/group) were primed with plasmid DNA encoding ISG15GG, ISG15AA, or empty vector in combination with plasmid DNA-ZIKV and boosted at week 22 with MVA-ZIKV. Ten days after the boost, spleens were obtained and ZIKV-specific responses were evaluated ex vivo by ICS after stimulation of splenocytes with a ZIKV E peptide pool, as described in Materials and Methods. (**A**) Magnitude of ZIKV E-specific CD4 (left panel) and CD8 (right panel) T cells. The values represent the sum of the percentages mean and SD of T cells producing CD107a and/or IFN-γ and/or TNF-α and/or IL-2 against the ZIKV E protein peptide pool. *P*-values were determined as described in Materials and Methods using an approach that corrects measurements for the medium response, calculating confidence intervals (**, *p* < 0.01; ***, *p* < 0.001). (**B**) Polyfunctional profile (based on expression of selected markers CD107a, IFNγ, TNFα, and IL-2) of total ZIKV E-specific CD4 (left panel) and CD8 (right panel) T-cell immune responses. Percentages represent the mean ± SD of ZIKV-specific CD4 (left) and CD8 (right) T cells producing one, two, three, or four cytokines. The response profiles are shown on the x-axis, and the percentages of T cells for each of the vaccination groups are shown on the y-axis. The pie charts summarize the proportion of S-specific T cells expressing one, two, three, or four functional markers, which each subset represented by a distinct color code. (**C**,**D**) ZIKV-specific Tfh-cell responses. (**C**) Magnitude of ZIKV-specific T CD4^+^CXCR5^+^PD-1^+^ (Tfh)-cell responses in splenocytes, assessed by ICS. Bars represent the mean ± SD of the total percentage of CD4^+^ Tfh cells expressing CD40L and/or producing IL-21 and/or IFN-γ following stimulation with ZIKV E protein plus ZIKV E peptide pool. Data are background (RPMI)-subtracted. Significant differences between immunization groups are indicated (***, *p* < 0.001). (**D**) Polyfunctional profile of ZIKV E-specific CD4^+^ Tfh cells, based on expression of CD40L, IL-21, and IFN-γ. Percentages represent the mean ± SD of ZIKV-specific CD4^+^ Tfh cells exhibiting one or more functional markers.

**Figure 5 vaccines-13-00696-f005:**
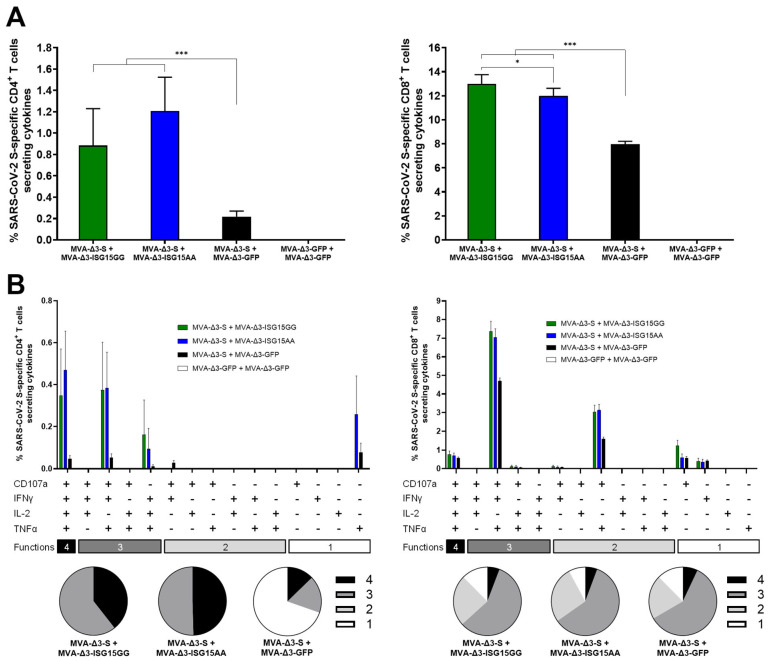
ISG15 co-administration improves SARS-CoV-2 S-specific T-cell responses in immunized mice. Female C57BL/6 mice (n = 5/group) were co-immunized at weeks 0 and 2 with MVA-Δ3-S + MVA-Δ3-GFP, MVA-Δ3-ISG15GG, or MVA-Δ3-ISG15AA. At 10 days after the boost, spleens were obtained and S-specific responses were evaluated by ICS after stimulation of splenocytes with an S peptide pool, as described in Materials and Methods. (**A**) Magnitude of S-specific CD4 (left panel) and CD8 (right panel) T cells. The values represent the sum of the percentages of the mean ± SD of T cells producing CD107a and/or IFN-γ and/or TNF-α and/or IL-2 against the SARS-CoV-2 S peptide pool. *P*-values were determined as described in Materials and Methods using an approach that corrects measurements for the medium response, calculating confidence intervals (*, *p* < 0.05; ***, *p* < 0.001). (**B**) Polyfunctional profile of S-specific CD4 (left panel) and CD8 (right panel) T-cell responses, based on the expression of CD107a, IFNγ, TNFα, and IL-2. Bars represent the mean ± SD of the percentage of S-specific CD4^+^ (left) and CD8 (right) T cells producing one, two, three, or four of these effector molecules. The x-axis indicates the combination of functions, while the y-axis shows the percentage of responding T cells for each immunization group. Pie charts illustrate the distribution of S-specific T cells exhibiting one, two, three, or four functions, with each category represented by a distinct color code.

## Data Availability

The raw data supporting the conclusions of this article will be made available by the authors, without undue reservation.

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
