# Peer review of "ISG15 as a Potent Immune Adjuvant in MVA-Based Vaccines Against Zika Virus and SARS-CoV-2"

_vaccines, 2025, doi:10.3390/vaccines13070696_

Round 1

Reviewer 1 Report

Comments and Suggestions for Authors

The authors assessed the immune-adjuvant function of ISG15 in MVA-based vaccines targeting Zika virus and SARS-CoV-2. They demonstrated that ISG15 serves as a promising immunomodulatory adjuvant, capable of enhancing both innate and adaptive immunity through viral vaccine platforms.

  1. The author concluded that the platform-dependent modulation of early innate immunity by ISG15 variants is influenced by differences in sample collection timing. Specifically, samples were collected 24 hours post-infection for MVA-based ISG15 and on day 4 post-injection for DNA-based ISG15. Furthermore, while comparisons were made between ISG15AA and ISG15GG in the MVA-based platform, as well as between ISG15AA and ISG15AA-mut in the DNA-based platform, direct cross-platform comparisons are challenging due to variations in experimental parameters.
  2. ISG15 improves early innate immune cell recruitment and enhances both the magnitude and polyfunctionality of antigen-specific CD4 and CD8 T cell responses. However, the impact of ISG15 on antibody-mediated immune responses remains to be elucidated.
  3. In figure 3, it is difficult to observe a slight reduction in IFN-I expression between co-infections of MVA-Δ3-S or MVA-ZIKV with MVA-Δ3-SG15AA and co-infection of MVA-B with MVA-Δ3-ISG15AA.
  4. Line 46, the correct term should be co-infection rather than co-expression.

Author Response

REVIEWER 1

The authors assessed the immune-adjuvant function of ISG15 in MVA-based vaccines targeting Zika virus and SARS-CoV-2. They demonstrated that ISG15 serves as a promising immunomodulatory adjuvant, capable of enhancing both innate and adaptive immunity through viral vaccine platforms.

  1. The author concluded that the platform-dependent modulation of early innate immunity by ISG15 variants is influenced by differences in sample collection timing. Specifically, samples were collected 24 hours post-infection for MVA-based ISG15 and on day 4 post-injection for DNA-based ISG15. Furthermore, while comparisons were made between ISG15AA and ISG15GG in the MVA-based platform, as well as between ISG15AA and ISG15AA-mut in the DNA-based platform, direct cross-platform comparisons are challenging due to variations in experimental parameters.

Response: We thank the reviewer for this observation. Indeed, the comparison between MVA- and DNA-based vaccine platforms is limited by the different time points selected for sample collection (24 hours post-infection vs. 4 days post-injection, respectively). These time points were chosen based on the distinct kinetics of immune activation associated with each platform. We have now added a clarification to the Discussion section, indicating this limitation and highlighting the challenges of direct cross-platform comparisons due to intrinsic differences in vector biology and immune kinetics (Page 18, line 611-616).

  1. ISG15 improves early innate immune cell recruitment and enhances both the magnitude and polyfunctionality of antigen-specific CD4 and CD8 T cell responses. However, the impact of ISG15 on antibody-mediated immune responses remains to be elucidated.

Response: We agree with the reviewer that the effect of ISG15 on antibody-mediated responses remains to be fully elucidated. While our current study primarily focuses on cellular immune responses, we are actively conducting experiments to evaluate humoral immunity, including measurements of antigen-specific IgG titers and neutralization assays. Preliminary data indicate that ISG15 does not significantly affect antibody production. We have now acknowledged this as a limitation of the present study and highlighted ongoing and future investigations aimed at clarifying the mechanisms underlying this observation in the Discussion section (Page 19, lines 645-652).

  1. In figure 3, it is difficult to observe a slight reduction in IFN-I expression between co-infections of MVA-Δ3-S or MVA-ZIKV with MVA-Δ3-SG15AA and co-infection of MVA-B with MVA-Δ3-ISG15AA.

Response: We appreciate the reviewer’s feedback. The differences between the various co-infections (MVA-B, MVA-ZIKV and MVA-Δ3-S) and MVA-Δ3-ISG15AA are very small and may not be apparent. Therefore, we have now included the mean value data in the legend of Figure 3 (Page 13, lines 450-453).

  1. Line 46, the correct term should be co-infection rather than co-expression.

Response: Thank you for pointing this out. The term has been corrected from "co-expression" to "co-infection" as suggested.

Reviewer 2 Report

Comments and Suggestions for Authors

The article discusses the study of candidate vaccines based on a viral vector - a vaccine strain of a virus related to smallpox and cowpox, modified by the insertion of a gene that stimulates the production of IL-15.

In the part concerning the preclinical evaluation of new adjuvants: when using a new adjuvant, a separate study of its immunogenicity, toxicity and characteristics, as well as vaccine formulations, is carried out (WHO guidelines on non-clinical evaluation of vaccines, Annex 1, TRS No 927). That is, the relevant data should be presented which were examined in the study of the ISG15 adjuvant, and the authors should indicate the sources (links).

Regarding vector vaccines, a regulatory document has not yet been developed that regulates the criteria for preclinical studies, but there are additional recommendations that require additional tissue cross-reactivity tests in cell cultures of human organs and tissues, designed to identify the potential impact of the vector strain and identify possible risks of using the candidate vaccine in humans.

Parameters for assessing immunogenicity are selected by researchers independently according to the established levels of protection to the corresponding antigens and, if necessary, an assessment of cellular immunity is carried out (including memory B cells and NK cells). With regard to vector vaccines, it is important to assess the immune response to the vector itself, since serial administration of such drugs may be ineffective due to the risk of inducing formation of immunological tolerance to the carrier vector.

In this study the authors confirm a model in which the ISG15 adjuvant exerts different immunostimulatory effects depending on the delivery platform. It was shown that co-delivery of antigen with ISG15AA significantly enhanced antigen-specific T cell responses in vivo. In ZIKV vaccination models, DNA priming with ISG15AA folowed by MVA-ZIKV boosting elicited the strongest CD4 and CD8 T cell responses, with increased polyfunctionality. A similar enhancement was observed in the SARS-CoV-2 model (See lines 604-609).

However, the history of development of great amount of adjuvants demonstrated their minimal implementation in healthcare practice, due to the presence of reactogenic properties. Despite the fact that the described ISG15 has a pronounced adjuvant and immunomodulatory effect, the safety of its use in practice will be given priority and in this regard, there are some questions that should be reflected in the presentation of the material. In vivo analysis revealed different patterns of immune cell infiltration among the experimental groups (Figure 1). In muscles (Figure 1A), MVA-Δ3-ISG15AA (group 2) caused a more pronounced (line 351) decrease in eosinophils and neutrophils, while in DLN (Figure 1B), the concentration of these cells does not differ from the groups of vaccinated females MVA-Δ3-ISG15GG (group 1) and MVA-Δ3-GFP (group 3). Does it mean that a lower incidence of local reactions can be expected at the injection site in animals (and potentially in humans) vaccinated with MVA-Δ3-ISG15AA, while the incidence of systemic reactions will be similar to that in the comparison groups?

How confident are you that overexpression of ISG15, especially the unconjugated variant of ISG15AA, which results in increased production of proinflammatory cytokines (lines 453, 454) in macrophages, when co-delivered with vaccine antigens will not affect the safety of using such adjuvanted vaccines in clinical practice? DNA-mediated delivery of ISG15AA has been shown to markedly enhance both the magnitude and functional quality of CD4+ and CD8+ T cell responses specific to ZIKV (lines 520, 521) and SARS-CoV-2 (lines 575, 576) following immunization with MVA-ZIKV and MVA-SARS-CoV-2, respectively. Can we expect non-specific transient activation of CD4+ and CD8+ cells to other viruses in the post-vaccination period, which is also important in the prevention of various respiratory infections in the early post-vaccination period?

Author Response

REVIEWER 2

  1. The article discsses the study of candidate vaccines based on a viral vector - a vaccine strain of a virus related to smallpox and cowpox, modified by the insertion of a gene that stimulates the production of IL-15.

In the part concerning the preclinical evaluation of new adjuvants: when using a new adjuvant, a separate study of its immunogenicity, toxicity and characteristics, as well as vaccine formulations, is carried out (WHO guidelines on non-clinical evaluation of vaccines, Annex 1, TRS No 927). That is, the relevant data should be presented which were examined in the study of the ISG15 adjuvant, and the authors should indicate the sources (links).

Response: We thank Reviewer 2 for this important comment. In response, we have now included a reference to the WHO guidelines on the non-clinical evaluation of vaccine adjuvants (Annex 1, TRS No. 927) (Page 18, lines 604-608).

  1. Regarding vector vaccines, a regulatory document has not yet been developed that regulates the criteria for preclinical studies, but there are additional recommendations that require additional tissue cross-reactivity tests in cell cultures of human organs and tissues, designed to identify the potential impact of the vector strain and identify possible risks of using the candidate vaccine in humans.

Response: We acknowledge the importance of cross-reactivity testing in human tissues. While our study did not specifically include human tissue cross-reactivity assays, we have clarified this in the revised manuscript and added this point as a limitation to be addressed in future translational studies. We have now addressed this observation in the Discussion section (Page 19, lines 6731-676).

  1. Parameters for assessing immunogenicity are selected by researchers independently according to the established levels of protection to the corresponding antigens and, if necessary, an assessment of cellular immunity is carried out (including memory B cells and NK cells). With regard to vector vaccines, it is important to assess the immune response to the vector itself, since serial administration of such drugs may be ineffective due to the risk of inducing formation of immunological tolerance to the carrier vector.

Response: As suggested, we have now included a discussion on the potential impact of anti-vector immunity, especially in the context of prime/boost regimens. While our current data show strong immune responses after heterologous prime/boost (DNA/MVA), we agree that repeated administration with MVA vectors could lead to vector-specific immune interference, which should be evaluated in longer-term studies. We have now addressed this observation in the Results section (Page 17, lines 561-537), Discussion section (Page 19, lines 661-664), and new Figure S2 illustrates the specific acute CD8 T cell immune response elicited against the MVA vector.

 4.In this study the authors confirm a model in which the ISG15 adjuvant exerts different immunostimulatory effects depending on the delivery platform. It was shown that co-delivery of antigen with ISG15AA significantly enhanced antigen-specific T cell responses in vivo. In ZIKV vaccination models, DNA priming with ISG15AA folowed by MVA-ZIKV boosting elicited the strongest CD4 and CD8 T cell responses, with increased polyfunctionality. A similar enhancement was observed in the SARS-CoV-2 model (See lines 604-609).

However, the history of development of great amount of adjuvants demonstrated their minimal implementation in healthcare practice, due to the presence of reactogenic properties. Despite the fact that the described ISG15 has a pronounced adjuvant and immunomodulatory effect, the safety of its use in practice will be given priority and in this regard, there are some questions that should be reflected in the presentation of the material. In vivo analysis revealed different patterns of immune cell infiltration among the experimental groups (Figure 1). In muscles (Figure 1A), MVA-Δ3-ISG15AA (group 2) caused a more pronounced (line 351) decrease in eosinophils and neutrophils, while in DLN (Figure 1B), the concentration of these cells does not differ from the groups of vaccinated females MVA-Δ3-ISG15GG (group 1) and MVA-Δ3-GFP (group 3). Does it mean that a lower incidence of local reactions can be expected at the injection site in animals (and potentially in humans) vaccinated with MVA-Δ3-ISG15AA, while the incidence of systemic reactions will be similar to that in the comparison groups?

Response: We appreciate this thoughtful observation. Our data indicate that vaccination with MVA-Δ3-ISG15AA is associated with reduced infiltration of eosinophils and neutrophils at the injection site, which may suggest a lower degree of local reactogenicity. This interpretation has been incorporated into the Discussion section (Pages 18-19, lines 626-629). However, while these findings are promising, additional studies are necessary to validate this observation and assess its relevance in clinical settings.

  1. How confident are you that overexpression of ISG15, especially the unconjugated variant of ISG15AA, which results in increased production of proinflammatory cytokines (lines 453, 454) in macrophages, when co-delivered with vaccine antigens will not affect the safety of using such adjuvanted vaccines in clinical practice?

Response: We appreciate the reviewer’s concern. While ISG15AA overexpression does lead to increased cytokine expression in macrophages, our in vivo data did not indicate overt signs of toxicity or systemic inflammation. We now elaborate on these findings in the Discussion section (Page 19, lines 667-670) and emphasize the need for detailed toxicological studies before clinical translation.

6.DNA-mediated delivery of ISG15AA has been shown to markedly enhance both the magnitude and functional quality of CD4+ and CD8+ T cell responses specific to ZIKV (lines 520, 521) and SARS-CoV-2 (lines 575, 576) following immunization with MVA-ZIKV and MVA-SARS-CoV-2, respectively. Can we expect non-specific transient activation of CD4+ and CD8+ cells to other viruses in the post-vaccination period, which is also important in the prevention of various respiratory infections in the early post-vaccination period?

Response: This is an excellent question. While ISG15AA enhances antigen-specific T cell responses, we did not observe non-specific activation of T cells in our model systems. However, we acknowledge that transient non-specific activation cannot be excluded and may, in fact, be beneficial in broadening early antiviral immunity. This has been discussed as a potential advantage and a topic for future investigation (Page 18, lines 589-592).